# Association between Cerebral Oxygen Saturation with Outcome in Cardiac Surgery: Brain as an Index Organ

**DOI:** 10.3390/jcm9030840

**Published:** 2020-03-19

**Authors:** Youn Yi Jo, Jae-Kwang Shim, Sarah Soh, Sungmin Suh, Young Lan Kwak

**Affiliations:** 1Department of Anesthesiology and Pain Medicine, Gil Hospital, Gachon University College of Medicine, 1198 Guwol-dong, Namdong-gu, Incheon 405-760, Korea; endless37@gilhospital.com; 2Department of Anesthesiology and Pain Medicine, Severance Hospital, Anesthesia and Pain Research Institute, Yonsei University College of Medicine, Yonsei-ro, Seodaemun-gu, Seoul 120-752, Korea; aneshim@yuhs.ac (J.-K.S.); yeonchoo@naver.com (S.S.); 3Department of Anesthesiology and Pain Medicine, Severance Hospital, Yonsei University College of Medicine, Yonsei-ro, Seodaemun-gu, Seoul 120-752, Korea; silvi37@naver.com

**Keywords:** cardiac surgery, cerebral desaturation, morbidity, regional cerebral oxygen saturation

## Abstract

While both baseline regional cerebral oxygen saturation (rSO_2_) and intraoperative rSO_2_ decreases have prognostic importance in cardiac surgery, evidence is limited in patients who received interventions to correct rSO_2_ decreases. The primary aim was to examine the association between rSO_2_ values (both baseline rSO_2_ and intraoperative decrease in rSO_2_) with the composite of morbidity endpoints. We retrospectively analyzed 356 cardiac surgical patients having continuously recorded data of intraoperative rSO_2_ values. Per institutional guidelines, patients received interventions to restore the rSO_2_ value to ≥80% of the baseline value. Analyzed rSO_2_ variables included baseline value, and area under the threshold below an absolute value of 50% (AUT50). Their association with outcome was analyzed with multivariable logistic regression. AUT50 (odds ratio, 1.05; 95% confidence interval; 1.01–1.08; *p* = 0.015) was shown to be an independent risk factor (along with age, chronic kidney disease, and cardiopulmonary bypass time) of adverse outcomes. In cardiac surgical patients who received interventions to correct decreases in rSO_2_, increased severity of intraoperative decrease in rSO_2_ as reflected by AUT below an absolute value of 50% was associated with a composite of adverse outcomes, implicating the importance of cerebral oximetry to monitor the brain as an index organ.

## 1. Introduction

Cerebral oximetry based on near-infrared spectroscopy measures regional tissue oxygen saturation (rSO_2_) at the watershed zone of the anterior and middle cerebral arteries [1]. Due to its non-invasiveness and the practicability of providing real-time rSO_2_ even during non-pulsatile perfusion, it has gained popularity as a key monitor in cardiac surgery over the past two decades.

Earlier studies were mainly focused on detecting adverse neurocognitive outcome and often showed conflicting results [2,3,4]. These inconsistencies were to be expected to some extent, considering the inability of cerebral oximetry to adequately assess the middle and posterior cerebral circulation. In addition, the relatively low incidence of stroke and diversity in the diagnosis of cognitive dysfunction posed significant challenges for studies in this regard [4,5].

With its increasing application in cardiac surgery, another use for cerebral oximetry has arisen—i.e., viewing the brain as an index organ for overall hemodynamic stability [6]. This seems logical considering that cerebral oximetry measures tissue oxygen saturation at the ischemia-vulnerable watershed zone while our physiologic response aims to provide cerebral perfusion even at the cost of systemic hypoperfusion [6]. In addition, approximately 70% of the measured oxygen saturation comes from the hemoglobin (Hb) of the venous bed, indicating that a decrease in rSO_2_ would more likely be the consequence of increased oxygen extraction, reflecting significant hemodynamic compromise assuming a constant cerebral metabolic rate [7].

In this context, two major areas of studies exist regarding the potential role of cerebral oximetry to monitor the brain as an index organ. First, studies elucidated the association between significant intraoperative reductions in rSO_2_ and adverse outcome, and whether interventions to correct these would improve prognosis [2,8]. Second, compelling evidence has also indicated the importance of low baseline rSO_2_ value concerning its association with adverse outcome [9]. However, being a potentially modifiable risk factor, efforts to correct decrease in rSO_2_ might alter the prognostic importance of baseline rSO_2_, whereas intraoperative decrease in rSO_2_ despite these efforts may show a meaningful association with adverse outcome. Yet, evidence to that regard is scarce.

In this retrospective review, the primary aim was to study the association between the rSO_2_ values (both baseline rSO_2_ and decrease in intraoperative rSO_2_) with overall outcome in cardiac surgical patients who received interventions based on cerebral oximetry monitoring. The secondary aim was to evaluate the variables that were associated with the rSO_2_ value that showed significant association with adverse outcomes.

## 2. Materials and Methods

### 2.1. Patients

After obtaining approval (20 December 2018) from the Institutional Review Board, we reviewed the medical records of patients ≥20 years who underwent cardiac surgery at the Severance Cardiovascular Hospital, Seoul, Korea, from 20 March 2017 to 10 April 2018. Of the 1082 records initially identified, we excluded patients having off-pump surgery, congenital heart disease operations, using hypothermic circulatory arrest, emergency operations, as well as patients with acute coronary syndrome, recent myocardial infarction, cerebrovascular disease that required surgery, and cerebrovascular events within the preceding one month. Accordingly, the records of 356 valvular heart surgery patients were analyzed (Figure 1).

### 2.2. rSO_2_ Measurement and Study Endpoints

The rSO_2_ was measured using the INVOS^TM^ Cerebral/Somatic Oximeter 5100 (Covidien, Dublin, Ireland) with bi-hemispheric near-infrared spectroscopy sensors. The baseline rSO_2_ value was obtained in the supine position in room air before anesthesia, and measurement was continued until intensive care unit (ICU) transfer. For the analysis of rSO_2_ values, after simultaneously obtaining both left and right rSO_2_ values, the lower value was selected for analysis. Excel 2016 (Microsoft Office, Redmond, WA, USA) was used to calculate the cumulative area under the threshold (AUT) of rSO_2_ values (80% of baseline value; and below an absolute value of 50%) [10]. AUT was calculated using the following formula: AUT (%∙min) = SUM {[rSO_2threshold_ – (rSO_2TN_ + rSO_2TN-1_)/2] * 0.5}, where, rSO_2TN_ and rSO_2TN-1_ are two consecutive rSO_2_ values, and 0.5 min represents the time interval between two consecutive rSO_2_ values. If the average value of the last two consecutive values [(rSO_2TN_ + rSO_2TN-1_)/2] was below zero, the average was excluded from the SUM. Therefore, AUT should reflect the severity of decrease in rSO_2_ as it has both the magnitude and time components. Minimal and maximal rSO_2_ values and maximal degree of desaturation were also determined.

The primary aim of the study was to evaluate the association between rSO_2_ values (both baseline rSO_2_ and intraoperative decrease in rSO_2_) with a composite of morbidity endpoints. The composite of morbidity endpoints included in-hospital mortality combined with postoperative morbidity endpoints as defined by the Society of Thoracic Surgery (STS) database registry [11]. These included permanent stroke, renal failure (serum creatinine [Cr] ≥4.0 mg/dL with increase of ≥0.5 mg/dL or 3 times most recent preoperative creatinine level, newly required dialysis), prolonged ventilation (>24 h), deep sternal wound infection, and any cause of reoperation in the absence of an evident surgical bleeding focus, which were checked during the postoperative hospitalization.

The secondary aim was to evaluate the factors including the hemodynamic and arterial-blood gas variables, temperature, and Hb, that were associated with either low baseline rSO_2_ value or significant decreases in intraoperative rSO_2_, depending on their association with adverse outcomes.

### 2.3. Perioperative Data Assessment

Preoperative data assessment included demographic data, comorbidities (hypertension, diabetes mellitus, congestive heart failure [New York Heart Association classification 3 or 4], myocardial infarction, chronic kidney disease [CKD, glomerular filtration rate under 60 mL/min/1.73 m^2^ for more than 3 months], cerebrovascular disease, chronic obstructive lung disease), medications, European System for Cardiac Operative Risk Evaluation (EuroSCORE), left ventricular ejection fraction (LVEF), and Hb and Cr levels.

Intraoperative data assessment included the duration of cardiopulmonary bypass (CPB) and aortic cross clamp, fluid balance, packed erythrocytes (pRBCs) transfusion, urine output, and the use of norepinephrine, vasopressin, milrinone, and dobutamine. Hemodynamic and blood gas variables including mean arterial pressure (MAP), cardiac index (CI), mixed venous oxygen saturation (SvO_2_), central venous pressure (CVP), mean pulmonary arterial pressure (mPAP), PaO_2_, PaCO_2_, and Hb concentrations were recorded at predetermined time points (before anesthetic induction [baseline], 15 min after anesthetic induction, 15 min after starting CPB, 20 min after weaning from CPB, and after sternal closure). Postoperative data assessment included fluid balance, transfusion requirement, chest tube drainage for 24 h, as well as the use of norepinephrine, vasopressin, milrinone, and dobutamine.

### 2.4. Perioperative Management Including Interventions Guided by Decrease in rSO_2_

All patients were managed according to standardized institutional anesthetic and CPB practice guidelines. Briefly, standard monitoring included bispectral index (BIS; A-200 Bispectral Index^®^ score monitor; Aspect Medical System Inc., Norwood, MA, USA), rSO_2_, pulmonary artery catheter, and transesophageal echocardiography. Anesthesia was provided using sufentanil and sevoflurane and BIS was maintained at 40–60. MAP was maintained at 60–80 mmHg using norepinephrine first, and if the target MAP could not be maintained with escalating doses of norepinephrine (maximum of 0.3 μg/kg/min), vasopressin was added (2.4–4 unit/h). Milrinone was used in cases of LVEF <30%, right ventricular dysfunction, or pulmonary hypertension. During CPB, non-pulsatile perfusion was utilized at 2.0–2.5 L/min/m^2^ using a tepid temperature (32–34 °C) and alpha-stat management. pRBCs were transfused when Hb was <7.0 g/dL during CPB or <8.0 g/dL otherwise.

All patients received interventions according to a modified institutional guideline based on the algorithm of Denault et al. [12] when significant decrease in rSO_2_ occurred (defined as a 20% reduction from baseline values), which were as follows: checking neck and cannulae position, increasing end-tidal CO_2_ to approx. 40 mmHg (in the absence of pulmonary hypertension), increasing the dose of norepinephrine or addition of vasopressin to increase the MAP above 65 mmHg up to 80 mmHg, if necessary. In addition, milrinone and/or dobutamine to maintain a SvO_2_ of ≥60% and CI > 2.0 L/min/m^2^ (increase of flow rate to 2.5 L/min/m^2^ during CPB) was administered, while efforts to optimize preload was done using mini-fluid boluses in 100 mL aliquots. Lastly, when all interventions failed to restore the rSO_2_ value above 80% of the baseline, pRBCs transfusion was considered at Hb of 8–9 g/dL.

### 2.5. Statistical Analysis

Statistical analyses were performed using SPSS version 17 (SPSS, Inc., Chicago, IL, USA) and SAS (version 9.4, SAS Inc., Cary, NC, USA) and R package (version 3.4.3. http://www.R-project.org). Results were shown as mean (standard deviation [SD]) or number of patients (%). Sample size was not determined a priori.

For intergroup comparisons, continuous data were tested for normality using the Kolmogorov–Smirnov test and analyzed using the Mann–Whitney U test or independent *t*-test, as required. Categorical data were analyzed using Fisher’s exact test or the chi-square test. Intergroup comparisons of the serially assessed data, including hemodynamic and laboratory variables, were done using linear mixed models with an unstructured covariance matrix. The model included group, time, and group-by-time as fixed effects. When the interaction of group, time, and group-by-time of the variables showed statistical significance, post hoc analysis was carried out with Bonferroni correction for the adjustment of multiple comparisons.

For analysis of the primary endpoint, to confirm independent risk factors of the composite of morbidity endpoints, multivariable logistic regression analysis was performed. First, well known risk factors were chosen a priori to minimize the introduction selection bias, which included age, CKD, congestive heart failure, and CPB duration in order to abide the rule of ten [13,14,15]. In terms of the variables of interest, rSO_2_, they were tested for their probability on the composite of morbidity endpoints using the area under the receiver operating characteristic (AUROC) curve. Then, rSO_2_ variables with the highest AUROC were introduced to the regression analysis, which were baseline and AUT 50. An ROC curve was constructed for the identified rSO_2_ variables with prognostic importance.

For analysis of the secondary outcome, Pearson’s correlation analysis was used to verify the correlation between the factors influencing rSO_2_ values (perfusion pressure, Hb, PaCO_2_, CI, mPAP) and rSO_2_. The Pearson’s correlation coefficient (r) was demonstrated. In addition, intergroup analysis of the hemodynamic and arterial-blood gas analysis variables was performed between patients who exhibited AUT50 greater than 113.2 min·% or not. A *p* < 0.05 was deemed as statistically significant.

## 3. Results

The composite of morbidity endpoints occurred in 47 of the 356 (13.2%) patients that were analyzed. In those patients, permanent stroke occurred in 5 patients (1.4%), reoperation in 7 patients (1.9%), renal failure in 8 patients (2.2%), mechanical ventilation >24 h in 29 patients (8.1%), deep sternal wound infection in 4 patients (1.1%), and mortality in 1 patient (0.3%).

Patients who exhibited a composite of morbidity endpoints had a higher incidence of preoperative CKD, higher incidence of heart failure, higher EuroSCORE, lower LVEF, lower Hb and albumin levels, higher Cr levels, longer CPB time, higher incidences of intraoperative pRBCs transfusion, and milrinone and vasopressin requirements compared to those without morbidities (Table 1 and Table 2).

Representative rSO_2_ values are shown in Table 3. All of the assessed rSO_2_ values including the baseline, AUT < 80% of baseline rSO_2_, and AUT < absolute value of 50% were significantly lower in the group with morbidity compared with the non-morbidity group, except for the percentage of maximal decrease from baseline rSO_2_ recovery to more than 80% of the baseline value at the end of the surgery (which occurred in 67% of patients). Subgroup analysis between patients whose rSO_2_ value was recovered to more than 80% of the baseline value at the end of the surgery and those who did not revealed no significant differences in the composite of postoperative morbidity endpoints (data not shown).

In the multivariable logistic regression analysis for finding risk factors of the composite of morbidity endpoints, age, preoperative CKD, AUT of rSO_2_ below an absolute value of 50%, and duration of CPB were identified as independent risk factors, whereas baseline rSO_2_ and AUT below 80% of the baseline rSO_2_ were not (Table 4). The cut-off value of AUT 50 to predict the composite of morbidity endpoints was 113.2 min·% with an AUROC of 0.697 (95% confidence interval, 0.607–0.787; *p* < 0.001) with 80.9% sensitivity and 40.8% specificity.

In Pearson correlation analysis, changes in MAP (r = 0.126; *p* < 0.001), Hb (r = 0.432; *p* < 0.001), CI (r = 0.110; *p* < 0.001), and CVP (r = −0.115; *p* < 0.001) were statistically significantly, but weakly correlated with changes in rSO_2_. In further analyses to find variables related to decrease in rSO_2_, there were no significant differences in MAP, CI, PaO_2_, CVP, and body temperature between patients who exhibited AUT of greater than 113.2 min·% below an absolute value of 50% (high AUT 50) or not (low AUT 50), while Hb levels were significantly lower and PaCO_2_ levels were significantly higher in the high AUT50 group throughout surgery (Table 5). Patients in the high AUT 50 group received more intraoperative pRBCs transfusion (58% versus 22%; *p* < 0.001) and had higher vasopressin requirements (77% versus 64%; *p* = 0.01) than patients in the low AUT 50 group.

## 4. Discussion

In this retrospective inquiry into the impact of baseline rSO_2_ and intraoperative decrease in rSO_2_ on a composite of morbidity endpoints in cardiac surgical patients who received interventions to correct decrease in rSO_2_ of more than 20% from baseline value, the AUT of rSO_2_ below an absolute value of 50% was identified as an independent risk factor along with age, preoperative CKD, and duration of CPB.

Cumulating evidence indicated the value of rSO_2_ in reflecting overall hemodynamic status and thus, a relationship between a low rSO_2_ value and systemically impaired tissue perfusion [8]. Indeed, studies have shown promising results supporting this concept of monitoring the brain as an index organ and have depicted associations of low baseline rSO_2_ values or prolonged intraoperative rSO_2_ desaturation with an increased risk of overall postoperative morbidity after cardiac surgery [6]. rSO_2_ has also been shown to predict abnormal cardiac function better than hemodynamic variables derived from a pulmonary artery catheter [16]. Although the evidence is not conclusive, further studies have shown the beneficial influence of interventions to correct intraoperative decrease in rSO_2_ on outcome [8,17]. As these corrective interventions mainly target improved perfusion [12], they have gained increased acceptance in our daily clinical practice. Consequently, by being a potentially modifiable risk factor, the value of baseline rSO_2_ as an aid for accurate risk stratification may be altered by intraoperative interventions to correct its decrease. In addition, evidence is also limited in terms of the predictive value of intraoperative decrease in rSO_2_ on outcome when being exposed to corrective interventions.

As our findings show, various assessed rSO_2_ values including baseline, minimal (the lowest measured value), AUT of 80% of the baseline, and AUT below an absolute value of 50%, were all significantly different between the morbid and non-morbid groups, indicating the prognostic importance of cerebral oximetry monitoring. However, the prognostic value of low baseline rSO_2_ was not present after adjusting for potential confounders of adverse outcome. In contrast, the severity of intraoperative decrease in rSO_2_ below an absolute value of 50%, expressed as AUT 50, was revealed to be an independent risk factor of adverse outcome together with advanced age, preoperative CKD, and prolonged CPB time. The assessed major STS morbidity endpoints are well validated surrogate measures of overall outcome [11]. In the current study, prolonged ventilation was most prevalent among the assessed morbidity endpoints. As with any other morbidity endpoints, prolonged ventilation is attributable to multiple, complex factors that cannot be properly delved in the current study. However, common to all morbidity endpoints, it is well-known that lung injury after cardiac surgery is related to the degree of hypoperfusion or ischemia-reperfusion injury during CPB [18]. In addition, hemodynamic instability would play a major role in prolonged ventilation as well, supporting the role of monitoring the brain as an index organ.

In previous studies, evidence was more solid in terms of the prognostic importance of low baseline rSO_2_ values for a hard, clinical endpoint, mortality, when compared to those of intraoperative decrease in rSO_2_. This evidence includes a prospective, observational study involving more than 1100 cardiac surgical patients, showing a close correlation between baseline rSO_2_ value and mortality, which was even more predictive than the EuroSCORE in high-risk patients [9]. Another retrospective study involving more than 2000 cardiac surgical patients also showed that a baseline rSO_2_ value was a simple and useful predictor of mortality [19]. It is difficult to deduct any plausible explanation for the discrepancy with our result as only one of the previous studies was observational [9], while the other retrospective study stated that rSO_2_ values were actively treated without any specific information regarding the cut-off value upon which treatment was initiated [19]. Yet, the discrepancy may be attributable to the fact that all patients received intraoperative interventions to correct the decrease in rSO_2_ value below a specific point (20% reduction from baseline) in the current study. Also, it may be related to the low mean baseline rSO_2_ values of the current study (58% versus >62%) when compared to other studies or to the suggested normal range of approximately 70% [9,20,21]. This would simply indicate a poorer baseline cardiovascular functional reserve, and thus a generally more guarded prognosis in the patients we studied.

Likewise, the low baseline values may have affected the results that showed only an intraoperative decrease in rSO_2_ below an absolute value of 50% was a meaningful predictor of outcome, whereas desaturation to 80% of the baseline value was not. The currently used cerebral oximetry device (INVOSTM Cerebral/Somatic Oximeter 5100) uses 2 wavelengths to detect rSO_2_ and accordingly, the U.S. Food and Drug Administration recommends its use only as a trend monitor. Thus, a decrease to 80% of the baseline rSO_2_ value was a commonly adopted threshold for interventions in many previous studies [17,22]. Yet, in patients with low baseline values, as in our studied population, 80% of the baseline would be below the meaningful cut-off found in our study, an absolute value of 50%. Various cut-off values were previously suggested to be related to adverse outcomes and were recommended as potential intervention thresholds, including the manufacturer’s suggestion of 40% (absolute value) [20]. As our results indicate, a cut-off of an absolute value of 50% may be an appropriate target for intervention in cardiac surgical patients with limited cardiovascular functional reserves, as reflected by low baseline rSO_2_ values in whom a 20% decrease from baseline would be even lower.

Variables that are associated with changes in intraoperative rSO_2_ and that are commonly targeted for cerebral oximetry-based interventions such as MAP, CI, CVP, and Hb, all showed significant correlation, reaffirming the importance of the brain as an index organ for overall hemodynamic status. In addition, patients in the high AUT 50 group received more pRBCs transfusion and vasopressin, and had higher PaCO_2_ levels and lower Hb levels compared to the low AUT 50 group, indicating that they apparently received more interventions to increase rSO_2_. Still, only 67% of the patients showed restoration of rSO_2_ value above 80% of the baseline value and considerable numbers of patients with AUT 50 were present in the morbid group despite corrective interventions. Studies have shown the lower limit of cerebral autoregulation to be higher than 90 mmHg in some cardiac surgical patients [23]. Additionally, cumulating evidence favors the use of restrictive transfusion strategies in cardiac surgery with Hb thresholds of 7–8 g/dL [24], while cerebral oximetry has been shown to be a useful monitor to safely implement restrictive transfusion [25]. Nonetheless, it remains questionable whether increasing the MAP to near or above 90 mmHg or giving pRBCs transfusions at Hb levels of above 9 g/dL to correct decrease in rSO_2_ would result in improved outcomes considering the inherent technical limitations related to cerebral oximetry and its well-known high sensitivity but low specificity in detecting adverse outcome [6]. This could also be observed in our ROC analysis, showing the predictive power of AUT 50 on a composite of morbidity endpoints at 80.9% of sensitivity and 40.8% of specificity.

The current study is linked to limitations intrinsic to its retrospective nature and the technical aspects of the cerebral oximetry (INVOSTM Cerebral/Somatic Oximeter 5100) including the approximate depth of measurement of 2 cm [26], while the distance from the skin to the frontal cerebral cortex could be longer in elderly patients with cortical brain atrophy [27]. Second, although interventions to restore the rSO_2_ values were guided by our given institutional protocol, we were unable to check the rate of non-adherence to the protocol. Third, composite of morbidity endpoints could only be observed in 47 patients, which have limited comprehensive introduction of variables known to affect outcome and thus, limiting the statistical power of the study.

Still, the strength of the current study lies in that it provides primary evidence regarding the prognostic value of baseline rSO_2_ as well as intraoperative rSO_2_ desaturation on overall outcome in cardiac surgical patients who received interventions to correct intraoperative decrease in rSO_2_ using a comprehensive rSO_2_ data set collected at 30-s intervals.

In conclusion, in cardiac surgical patients who received interventions according to cerebral oximetry monitoring, increased severity of intraoperative decrease in rSO_2_ as reflected by AUT below an absolute value of 50% was found to be an independent risk factor of dismal prognosis, implicating an important ancillary role of cerebral oximetry to monitor the brain as an index organ.

## Figures and Tables

**Figure 1 jcm-09-00840-f001:**
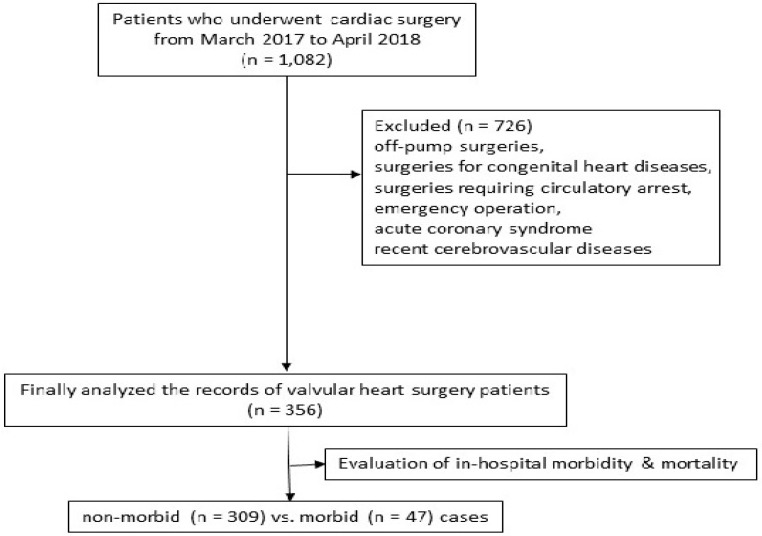
Flow chart.

**Table 1 jcm-09-00840-t001:** Patients’ characteristics and preoperative data.

	Non-Morbid (n = 309)	Morbid (n = 47)	*p-*Value
Age (years)	61.8 ± 13.3	67.9 ± 11.9	0.002
Female sex (n)	147 (48)	26 (55)	0.322
Body surface area (m^2^)	1.68 ± 0.20	1.64 ± 0.19	0.140
Preoperative morbidity (n)			
Hypertension	144 (47)	33 (70)	0.003
Diabetes mellitus	53 (17)	11 (23)	0.298
Congestive heart failure	57 (18)	15 (32)	0.032
CKD	27 (9)	17 (36)	<0.001
COPD	11 (4)	1 (2)	0.612
Cerebrovascular attack	34 (11)	5 (11)	0.941
EuroSCORE	5.6 ± 3.2	8.0 ± 3.0	<0.001
LVEF (%)	61.4 ± 12.2	57.3 ± 15.7	0.040
Preoperative medication (n)			
Beta blocker	119 (39)	14 (30)	0.468
Calcium channel blocker	78 (25)	17 (36)	0.115
ACE-I/ARB	144 (47)	23 (49)	0.765
Diuretics	218 (71)	34 (72)	0.801
Preoperative laboratory value			
Hemoglobin (g/dL)	12.8 ± 1.9	11.8 ± 2.3	0.001
Serum creatinine (mg/dL)	0.90 ± 0.71	1.53 ± 1.87	<0.001
Albumin (g/dL)	3.97 ± 0.45	3.69 ± 0.63	<0.001
Operations (n)			0.662
Mitral valve replacement	110 (36)	15 (32)	
Aortic valve replacement	142 (46)	20 (43)	
Double valve replacement	40 (13)	8 (17)	
Others	16 (4)	4 (9)	

Values are mean ± standard deviation, or number of patients (%). CKD, chronic kidney disease; COPD, chronic obstructive pulmonary disease; EuroSCORE, European system for cardiac operative risk evaluation; LVEF, left ventricular ejection fraction; ACE-I, angiotensin converting enzyme inhibitor; ARB, angiotensin receptor blocker.

**Table 2 jcm-09-00840-t002:** Intraoperative data.

	Non-Morbid (n = 309)	Morbid (n = 47)	*p-*Value
ACC duration (min)	78.7 ± 39.4	91.0 ± 46.5	0.090
CPB duration (min)	109.8 ± 47.6	127.5 ± 51.1	0.031
Total infused fluid (mL)	1090.2 ± 375.6	1192.6 ± 471.2	0.165
pRBCs transfusion (n)	125 (41)	32 (68)	<0.001
Total urine output (mL)	937.8 ± 519.4	731.5 ± 602.9	0.030
Norepinephrine (n)	307 (99)	47 (100)	0.580
Milrinone (n)	99 (32)	22 (47)	0.046
Vasopressin (n)	215 (70)	41 (87)	0.012
Dobutamine (n)	5 (2)	2 (4)	0.227

Values are mean ± standard deviation, or number of patients (%). ACC, aorta cross clamping; CPB, cardiopulmonary bypass, pRBCs, packed red blood cell.

**Table 3 jcm-09-00840-t003:** Representative values of regional cerebral oxygen saturation.

	Total (n = 356)	Non-Morbid (n = 309)	Morbid (n = 47)	*p-*Value
Baseline rSO_2_ (%)	58.0 ± 10.1	58.7 ± 9.7	52.8 ± 11.5	0.002
Maximal rSO_2_ (%)	70.8 ± 9.1	71.7 ± 8.7	65.0 ± 9.3	<0.001
Minimal rSO_2_ (%)	37.6 ± 9.9	38.4 ± 9.6	32.7 ± 10.4	0.001
AUT 80_base_ (min%)	355.2 ± 537.4	328.2 ± 479.1	532.4 ± 809.4	0.015
AUT 50 (min%)	678.7 ± 1046.5	566.6 ± 895.7	1415.6 ± 1562.0	<0.001
Complete recovery (n)	72 (20)	57 (19)	15 (32)	0.034
80% recovery (n)	240 (67)	208 (68)	32 (68)	0.988
Maximal decrease (%)	34.7 ± 14.6	34.2 ± 14.1	37.3 ± 17.4	0.176

Values are mean ± standard deviation or number of patients (%). rSO_2_, regional cerebral oxygen saturation; AUT 80base, area under the threshold below 80% of baseline rSO_2_; AUT50, area under the threshold below an absolute value of 50% of rSO_2_; complete recovery, restoration of rSO_2_ to the baseline value at the time of sternal closure; 80% recovery; restoration of rSO_2_ over the 80% of baseline value at the time of sternal closure. Maximal decrease; maximal decrease of rSO_2_ compared to the baseline value.

**Table 4 jcm-09-00840-t004:** Logistic regression analysis for finding risk factors of composite of morbidity end points.

	Univariable			Multivariable		
	OR	95% CI	*p-*Value	OR	95% CI	*p-*Value
Age (years)	1.04	1.01–1.08	0.004	1.05	1.02–1.09	0.004
Chronic kidney disease (n)	5.92	2.90–12.09	<0.001	4.92	2.13–11.36	<0.001
Congestive heart failure (n)	2.07	1.05–4.08	0.035	1.41	0.65–3.09	0.386
Baseline rSO_2_ (%)	0.95	0.92–0.98	<0.001	1.02	0.97–1.07	0.424
AUT 50 (min%)	1.06	1.03–1.08	<0.001	1.05	1.01–1.08	0.015
CPB duration (min)	1.93	1.09–3.39	0.023	1.01	1.00–1.02	0.014

OR, odds ratio; 95% CI, 95% confidence interval; rSO_2_, regional cerebral oxygen saturation; AUT50, area under the threshold below an absolute value of 50% of regional cerebral oxygen saturation; CPB; cardiopulmonary bypass.

**Table 5 jcm-09-00840-t005:** Analysis of variables related to rSO_2_ desaturation.

Variables	Group	Time				
		Baseline	Ind 15 min	CPB 15 min	PostCPB	Sternal Closure
MAP (mmHg)	High AUT50	90.9 ± 16.4	75.0 ± 10.5	63.7 ± 11.3	67.6 ± 9.9	74.9 ± 11.7
	Low AUT50	89.1 ± 15.0	74.9 ± 10.5	61.9 ± 13.0	68.9 ± 10.2	77.0 ± 12.8
CI (L/min/m^2^)	High AUT50		2.1 ± 0.7		2.5 ± 0.7	2.2 ± 0.6
	Low AUT50		2.2 ± 0.8		2.6 ± 0.9	2.2 ± 0.7
CVP (mmHg)	High AUT50		10.5 ± 3.6		10.6 ± 2.6	11.5 ± 2.8 *
	Low AUT50		9.8 ± 3.0		10.1 ± 2.8	10.6 ± 2.2
PaCO_2_ (mmHg)	High AUT50		34.2 ± 3.4 *	32.9 ± 3.9	34.4 ± 3.9	36.9 ± 3.5 *
	Low AUT50		33.5 ± 3.0	33.1 ± 3.8	33.8 ± 3.6	35.7 ± 3.3
PaO_2_ (mmHg)	High AUT50		176.0 ± 44.5	329.0 ± 57.6	185.2 ± 41.8	167.7 ± 42.9
	Low AUT50		179.0 ± 39.9	326.5 ± 59.4	185.6 ± 37.9	171.7 ± 44.6
Temperature (°C)	High AUT50		36.1 ± 0.6	32.6 ± 2.7	36.4 ± 0.8	36.4 ± 0.5
	Low AUT50		36.0 ± 0.6	32.2 ± 3.1	36.4 ± 0.4	36.4 ± 0.5
Hemoglobin (g/dL)	High AUT50	12.2 ± 2.3 *	11.5 ± 1.7 *	7.8 ± 1.2 *	8.2 ± 1.0 *	9.6 ± 1.0 *
	Low AUT50	13.2 ± 1.8	12.3 ± 1.5	8.4 ± 1.5	8.8 ± 1.2	10.0 ± 1.1

Values are mean (standard deviation). High and low AUT50, Patients who exhibited area under the threshold of regional cerebral oxygen saturation below and absolute value of 50% greater than 113.2 min% or not; Baseline, before anesthetic induction; Ind 15 min, 15 min after anesthetic induction; CPB 15 min, 15 min after staring cardiopulmonary bypass (CPB); PostCPB, 20 min after weaning from CPB; MAP, mean arterial pressure; CI, cardiac index; CVP, central venous pressure, * *p* < 0.05 vs. low AUT50.

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
