# Peer review of "Association between Cerebral Oxygen Saturation with Outcome in Cardiac Surgery: Brain as an Index Organ"

_jcm, 2020, doi:10.3390/jcm9030840_

Round 1

Reviewer 1 Report

one of the major limitations of cerebral Oxygen saturation is the age of patients, this because there is an increase in distance between brain ond bone; this is confirmed by many studies with CT brain scan

Please clarify this point or underline it in study limitation

Author Response

One of the major limitations of cerebral Oxygen saturation is the age of patients, this because there is an increase in distance between brain and bone; this is confirmed by many studies with CT brain scan

Answer: We acknowledge your concern regarding the depth of measurement of approximately 2 cm and the potential for an increase in the depth above 2 cm in elderly patients with cortical brain atrophy. We have added this as a limitation as you commented (P9 L 308-310): ‘The current study is linked to limitations intrinsic to its retrospective nature and the technical aspects of the cerebral oximetry (INVOSTM Cerebral/Somatic Oximeter 5100) including the approximate depth of measurement of 2 cm [21], while the distance from the skin to the frontal cerebral cortex could be longer in elderly patients with cortical brain atrophy [27].'

Thank you for your constructive comment on our manuscript.

Reviewer 2 Report

Summary:

Thank you for giving me the opportunity to review the manuscript titled: “Association between cerebral oxygen saturation with outcome in cardiac surgery: brain as an index organ”. In this retrospective study, the authors reported that AUT50 of rSO2 was independently associated with adverse outcomes in cardiac surgical patients who received interventions to correct decreases in rSO2.

The reviewer think that the strengths of this study include the fact that the reduction of rSO2 was managed according to the standardized protocol. On the other hand, the main limitation of this study is that only few of confounders were adjusted because of small number of outcomes.

Critiques:

  1. abstract: AUT should be spelled out when it first appears.
  2. methods: page 3 lane 81: “the lowest value of either side was selected for analysis.”: Do you mean that the one with the lower baseline value of the left or right rSO2 was used for analysis?
  3. methods: How long is the follow-up period to assess morbidity endpoint?
  4. method: Of the many potential confounders, only a few have been adjusted to assess independent relationship between rSO2 and outcomes. Please provide rationales to for choosing these factors. It is desirable to provide literature.
  5. methods: Definitions of comorbidities are not provided. Definitions of CKD and CHF are important because these variables are used for multivariable analysis to assess the association between rSO2 and outcomes.
  6. methods: What type of monitor did you use to optimize preload?
  7. methods: How did you determined the threshold value of 50%?
  8. results: Most morbidities are prolonged mechanical ventilation. How do you think low rSO2 affect duration of mechanical ventilation?

Author Response

To the comments of the Reviewer #2

abstract: AUT should be spelled out when it first appears.

Answer: We have corrected it as you commented (P1 L23).

methods: page 3 lane 81: “the lowest value of either side was selected for analysis.”: Do you mean that the one with the lower baseline value of the left or right rSO2 was used for analysis?

Answer: Yes. To avoid any confusion, we have simplified the sentence to: ‘after simultaneously obtaining both left and right rSO2 values, the lower value was selected for analysis’ (P3 L81)

methods: How long is the follow-up period to assess morbidity endpoint?

Answer: As with in-hospital mortality, the follow-up period for the morbidity endpoints were confined to ‘during the postoperative hospitalization’ period. We have added this to the Methods section to clarify this matter as you commented (P3 L97)

method: Of the many potential confounders, only a few have been adjusted to assess independent relationship between rSO2 and outcomes. Please provide rationales to for choosing these factors. It is desirable to provide literature.

Answer: Upon preparing the manuscript, we had consulted our institutional statistical team that selection of variables for multivariable analysis would be best if they were chosen a priori to minimize the introduction of selection bias. However, the composite of morbidity/mortality endpoint occurred only in 47 patients, which limited the number of variables to 5 not to over fit the multivariable model. Thus, we have chosen age, chronic kidney disease, congestive heart failure, and CPB duration as these were most consistently proven risk factors from previous studies [Lee TH. Circulation 1999;100:1043-9., Salis S. J Cardiothorac Vasc Anesth 2008;22:814-22., Najafi M. World J Cardiol 2014;6:1006-21], while we also have to introduce our variable of interest, rSO2. Nonetheless, we acknowledge your concern regarding the incomprehensive introduction of known risk factors, such as diabetes, COPD, and cerebrovascular accidents…etc. due to the low incidence of composite of morbidity endpoints (incidentally, the incidence of diabetes, COPD, and cerebrovascular accidents did not differ between patients who exhibited composite of morbidity endpoints or not, while age, CKD, CHF, and duration of CPB did show significant intergroup differences [Table 1 and 2]). We have clearly stated this as a limitation of our study in our first draft (P9 L312-314 ). We have also stated the rationale and statistical background for choosing those variables a priori in the statistical analysis section (P4 L152-160 ) and we have added relevant references validating the chosen variables as major risk factors to clarify this matter as you commented [Lee TH. Circulation 1999;100:1043-9., Salis S. J Cardiothorac Vasc Anesth 2008;22:814-22., Najafi M. World J Cardiol 2014;6:1006-21] (P9 L356 – P10 L361 ).

methods: Definitions of comorbidities are not provided. Definitions of CKD and CHF are important because these variables are used for multivariable analysis to assess the association between rSO2 and outcomes.

Answer: CKD was defined as glomerular filtration rate under 60 ml/min/1.73 m2 for more than 3 months (KDIGO guideline) and congestive heart failure was defined as New York Heart Association classification 3 or 4. We have added these definitions to the Methods section to clarify this matter as you commented (P3 L103-105).

methods: What type of monitor did you use to optimize preload?

Answer: All patients were monitored with pulmonary artery catheter and TEE during surgery. PPV or SVV is impractical to use in valvular heart surgery as they cannot be used during open chest condition or rhythm other than sinus. Thus, we usually use fluid challenge in aliquots of 200-300 ml when the mean arterial pressure is below 60 mmHg and the PCWP is below 12 mmHg simultaneously, while the cardiac index is below 2 L/min/m2 or SvO2 is below 60%. Then we reassess all the variables again. We also use the TEE to rule out possible contractile dysfunction or other abnormalities that need to be corrected. We also look at the position of the interatrial septum and interventricular septum to rule out extreme forms of congestion. We would be happy to provide this information in detail in the Methods section. Yet, we did not do it not to give too much detail, as all patients were managed the same according to a standardized institutional anesthetic and CPB practice guidelines as we mentioned in the Methods section (P3 L118-121).

methods: How did you determined the threshold value of 50%?

Answer: Many studies showed meaningful cut-offs in absolute values of 60% or 50%. We have chosen 50% as a cut off value according to previous literatures considering that mean baseline rSO2 values were relatively low (52-58%) [J Cardiothorac Vasc Anesth 2011;25:95-104]. We have added this reference to the corresponding Methods section to clarify this matter (P3 L83). When introducing to the multivariable model, AUT 50 showed the highest AUROC among the rSO2 values as we mentioned in the statistical analysis section (P4 L155-160).

results: Most morbidities are prolonged mechanical ventilation. How do you think low rSO2 affect duration of mechanical ventilation?

Answer: Prolonged ventilation is a well-acknowledged morbidity endpoint by the STS (Society of Thoracic Surgeons), which is attributable to multiple factors. It would be influenced by the patients’ preoperative lung function, cardiac function, renal function…etc. In addition, the degree of hypoperfusion / ischemia-reperfusion injury during cardiopulmonary bypass is considered to be one of the major factors related to the lung injury as the lungs would only receive blood supply from the bronchial artery leaving the respiratory bronchioles (alveoli) subject to ischemic insult during CPB with or without cardioplegic arrest. In that context, many of the above-mentioned major causes would be dependent on the overall hemodynamic status, and therefore, one can speculate that viewing the brain as an index organ could be an explanation to the observed results of our study. Of course, one cannot attribute low rSO2 as the sole or major cause of the assessed morbidity endpoints including prolonged ventilation as other risk factors such as age, CKD, and CPB duration were also found to be independent risk factors as well. Still, our findings clearly show that monitoring the brain as an index organ by way of cerebral oximetry has an additive value for risk stratification along with other well-known risk factors of adverse outcome. The composite of major morbidity endpoints proposed by the STS is a well-validated surrogate measure of overall outcome, and we think that it would be beyond the scope to describe detailed possible causes of each individual morbidity endpoints. We have carefully added the following comments to the Discussion section to clarify this matter as you commented (P7 L252- P L256 ): ‘The assessed major STS morbidity endpoints are well validated surrogate measures of overall outcome. In the current study, prolonged ventilation was most prevalent among the assessed morbidity endpoints. As with any other morbidity endpoints, prolonged ventilation is attributable to multiple complex factors that cannot be properly delved in the current study. However, common to all morbidity endpoints, it is well-known that lung injury after cardiac surgery is related to the degree of hypoperfusion or ischemia-reperfusion injury during CPB [Best Practice & Research Clinical Anaesthesiology 29 (2015) 163-175]. Also, hemodynamic instability would play a major role in prolonged ventilation as well, supporting the role of monitoring the brain as an index organ.’

Thank you for your constructive comments.

Round 2

Reviewer 2 Report

The reviewer think that the manuscript has clearly improved after revision and the authors have been able to answer the queries addressed.